# Substitute for polyethylene (PE) films: A novel cow dung-based liquid mulch on silage cornfields

Xiangjun Yang[1,2‡], Lu Li[1‡], Wuyun Zhao[1]*, Xuan Li[3], Yongsong Mu[4], Maohan Chen[2], Xiaoqiang Wu[2]

**1** Faculty of Mechanical and Electrical Engineering, Gansu Agricultural University, Lanzhou, China, **2** Faculty of Mechanical Engineering, Chengdu University, Chengdu, China, **3** Faculty of Water Conservancy and Hydropower Engineering, Gansu Agricultural University, Lanzhou, China, **4** Huarui Agricultural Company, Zhangye, China

‡ XY and LL are co-first authors of this work.
* zhaowy@gsau.edu.cn

**Data Availability Statement:** All relevant data are within the paper and its Supporting information files.

**Funding:** Lu Li acknowledges the supports by the Gansu Province Natural Science Foundation for the

## Abstract

To prevent soil pollution caused by polyethylene (PE) films in the central region of Gansu, China, liquid mulching made from cow dung (CDLM) was trailed in silage maize fields. The degradation of CDLM and PE films, soil temperature, soil organic matter content, silage maize yield and water use efficiency (WUE) were evaluated for three years (2018–2020). The degradability of CDLM has been found to be much stronger than the one of PE films, with CDLM degrading 40–60 days after sowing and finishing around 100 days. CDLM had a lower insulating impact than PE films but a higher insulating effect than non-mulching films as the control (CK); CDLM could successfully increase soil organic matter, with a total increase of 1.01% over three years. CDLM increased silage maize yield by 6.2% compared to PE films and 17.2% compared to CK. Consequently, CDLM may be an interesting alternative to PE films for enhancing silage maize yield while decreasing soil contamination.

## Introduction

The mulching technique is widely used in agriculture to ensure and enhance crop yields, especially in locations with little rainfall and cold weather. The polyethylene (PE) films, which are petroleum-based products, are effective in the conservation of soil water, regulation of soil temperature, control of the weed, and improvements in water use efficiency (WUE) and yield [1–3]. However, despite these positive effects, PE films do not readily biodegrade and are even non-degradable [4–7]. After utilization, the residual plastic segments of the PE films will cause detrimental soil pollution problems. The residual plastic segments will reduce soil permeability and impede water and nutrients absorption resulting in crop yield reduction [8–10]. Moreover, due to the low recovery rate of used-PE films in north of China, the amount of residual PE films per hectare has now reached a new level of ~60–349 kg [11], which threatens the food safety and soil quality [12]. Furthermore, the release of plastic fragments from the residual PE

Youth (20JR10RA554) and GAU-KYQD-2019-16. The funders had no role in study design, data collection and analysis, decision to publish, or preparation of the manuscript.

**Competing interests:** The authors have declared that no competing interests exist.

films in the fields has been recognized as a primary source of plastic accumulation in the sea [13]. Hence, the "white revolution" which was brought from the PE films has reversed to "white pollution" for modern agriculture [14].

Herein above statements, many efforts have been made to synthesis various types of degradable mulching films to try to supersede PE films to lessen environmental pollutions [15]. Biodegradable plastic mulching films includes synthetic biodegradable plastics and natural polymer blend biodegradable plastics. The synthetic biodegradable plastics are prepared from polycaprolactone (PCL) [16], polylactic acid (PLA) [17] and poly adipic acid/butylene terephthalate (PBAT) [18]. The natural polymer blend biodegradable plastics are synthesized by blending starch, chitin, cellulose and chitosan with PE, polyvinyl alcohol (PVA), polyester, etc. [19–21]. Though the performance of biodegradable plastic mulching films is close to the PE films, these materials are mostly chemically synthesized, and the effects on the soil after being degraded should be further verified. Straw mulches were once suggested as an alternative to PE films because they can inhibit weed growth effectively, can be degraded into organic substances without recovery, and have almost no adverse effects on soil [22, 23]. But plant fiber mulches have lower insulation and mechanical qualities than PE films and also deteriorate swiftly into pieces and are blown away after being exposed to rain or wind within weeks [24–26].

Along with material advancements, nowadays, much attentions have been taken on the way of mulch laying. In agriculture, the liquid film is another new mulching film, which can be laid efficiently via spraying. This spray method would produce an artificial coating on the surface of plants to avoid insects, diseases and water transpiration. Several sprayable biodegradable solutions for soil mulching have been developed by utilizing natural polymers such as starch, cellulose, chitosan, alginate and glucomannan [27]. Chiellini developed a sprayable film by using waste gelatine and PVA mixes including sugar cane bagasse [28]. Adhikari recently reported a new sprayable biodegradable polymer membrane, water dispersed, which can be used in irrigated systems to prevent evaporating [29].

Cow dung, as waste of animal husbandry, is another primary pollution in rural areas and has become an urgent problem to solve. However, cow dung is not only a source of pollution but also a recyclable resource which attracts broad attention. Nowadays, it can be seen in widespread applications including farmland manure, clean energy [30, 31], sewage treatment [32, 33] and composite materials [34]. In addition, soil covered with cow dung, because of the biological fermentation, exhibits more microorganism species, which improve soil fertility as well as soil properties [35–40].

Zhangye City (locates in Gansu Province, the northwest of China) has a typical temperate continental climate, which is mainly manifested in significant changes in annual and daily temperature. It is cold in winter and hot in summer, and with less precipitation throughout the year. Meanwhile, Zhangye City is one of the largest corn seed production bases in China, and animal husbandry is another booming enterprise with almost 600,000 heads of castles. While benefiting to local economic incomes, its climate requires that a large amount of PE films have to be used before corn sowing. At the same time, animal husbandry generates waste disposal problem, with approximately $6.48 \times 10^6$ tons of cow dung per year [41]. Thus, rational development and utilization of waste to solve the "white pollution" brought by PE films are of significant ecological interest.

In this work, we produced CDLM from cow dung and then sprayed it to silage maize fields after seeding of crops. A three-year field study was conducted to investigate the degradation and insulating properties of CDLM, the effect of CDLM on the soil organic matter (SOM) content at a depth of 10 cm, and the impact of CDLM on the water use efficiency (WUE) and yield of the crop.

## Methods and materials

### Site description

The field experiment was carried out during the 2014–2016 maize growing seasons at the planting and breeding base (100 625′N, 38 375′E, 1673 m ASL) of Huarui Agricultural Company, Eco-industrial Park, Minle County, Zhangye City, Gansu Province, China. Each experimental plot was 0.6 ha. The regional groundwater table remained at a depth of 8 m. Initial soil physical and chemical properties in 0–20 cm soil layer were: SOM content 10.9 g·kg$^{-1}$, pH = 8.38, with a mean 14.5% field water holding capacity, total N, P, and available K was 0.034 g·kg$^{-1}$, 0.03 g·kg$^{-1}$, and 0.15 g·kg$^{-1}$, respectively. The site practiced mulched drip irrigation since 2015. The mean soil bulk density of the topsoil (0–20 cm depth) was 1.488 g·cm$^{-3}$.

This region has an arid continental climate. The mean annual sunshine duration is 3000 h, and the frost-free period lasts for 140 days a year. The accumulated temperatures above 10˚C and 15˚C were 3169˚C and 2595˚C in 2018, 3138.4˚C and 2538.4˚C in 2019, and 3176.2˚C and 2584.1˚C in 2020, respectively. As shown in Fig 1, the precipitations and mean daily temperatures of the experimental sites during the cropping season (April to September) were 242.2 mm and 17.4˚C in 2018, 244.1 mm and 17.5˚C in 2019, and 222.3 mm and 17.5˚C in 2020, respectively.

### Preparation and spraying of CDLM on the plots

The preparation process and spraying CDLM on the plots are shown in Fig 2. The first step was to flush the cow dung in the enclosure into the settling pond, and to collect the liquid dung with a sand content lower than 2% through multi-stage sand settling and impurity-removing treatments. The second step was to separate the liquid from the cow dung to obtain the cow dung with a moisture content about 70%. For grinding pulp, the separated cow dung was fermented by anaerobic and aerobic fermentations to reduce the Chemical Oxygen Demand (COD), nitrogen, and fecal coliforms on supernatant to 2500 ± 500 mg•L$^{-1}$, 550 ± 75 mg•L$^{-1}$, and ~ 358 mg•L$^{-1}$, respectively. In the third step, the pulped cow dung was mixed with crushed wheat straw in a ratio of 3:2, and water was added to a solid-liquid ratio of 1:9. After sowing, the configured CDLM was transported by a transport vehicle (20 tons capacity) and sprayed on the field (60 tons per hectare).

### The design of field experiments

Three treatments were experimented with: (i) PE films, (ii) CDLM and (iii) CK. Green organic cropping patterns and mechanical harvesting methods were employed. The experiment had a random block design, with a plot area of 0.2 ha and three repetitions, 90kg•ha$^{-1}$ of PE films and 60 tons•ha$^{-1}$ of CLDM. Variety: Long silage No.1 was selected. The plant spacing was 20 cm, the narrow row spacing was 25 cm, and the wide row spacing was 75 cm, as shown in Fig 3. We sowed 10000 holes per 0.1 ha, preserving 9000 plants to guarantee the emergence rate. Drip irrigation under film was used in the experiment. The flow of drip irrigation was 3 L•h$^{-1}$, and the whole irrigation amount was 3750 m$^3$•ha$^{-1}$. Urea (CO(NH$_2$)$_2$ with N content of 46%), monopotassium phosphate (KH$_2$PO$_4$, with P$_2$O$_5$ content of 52%) and KCl (with K of 57%) were applied via drip irrigation laterals. Detailed information on irrigation scheduling at different silage corn growth stages is shown in Table 1.

Drip irrigation laterals with the type of Single wing labyrinth (Tianye Co. Ltd., Xinjiang, China) were employed. The external diameter and wall thickness are 16 mm and 0.3 mm, respectively. Interval spacing between two emitters is 30 cm. A water pump was installed to

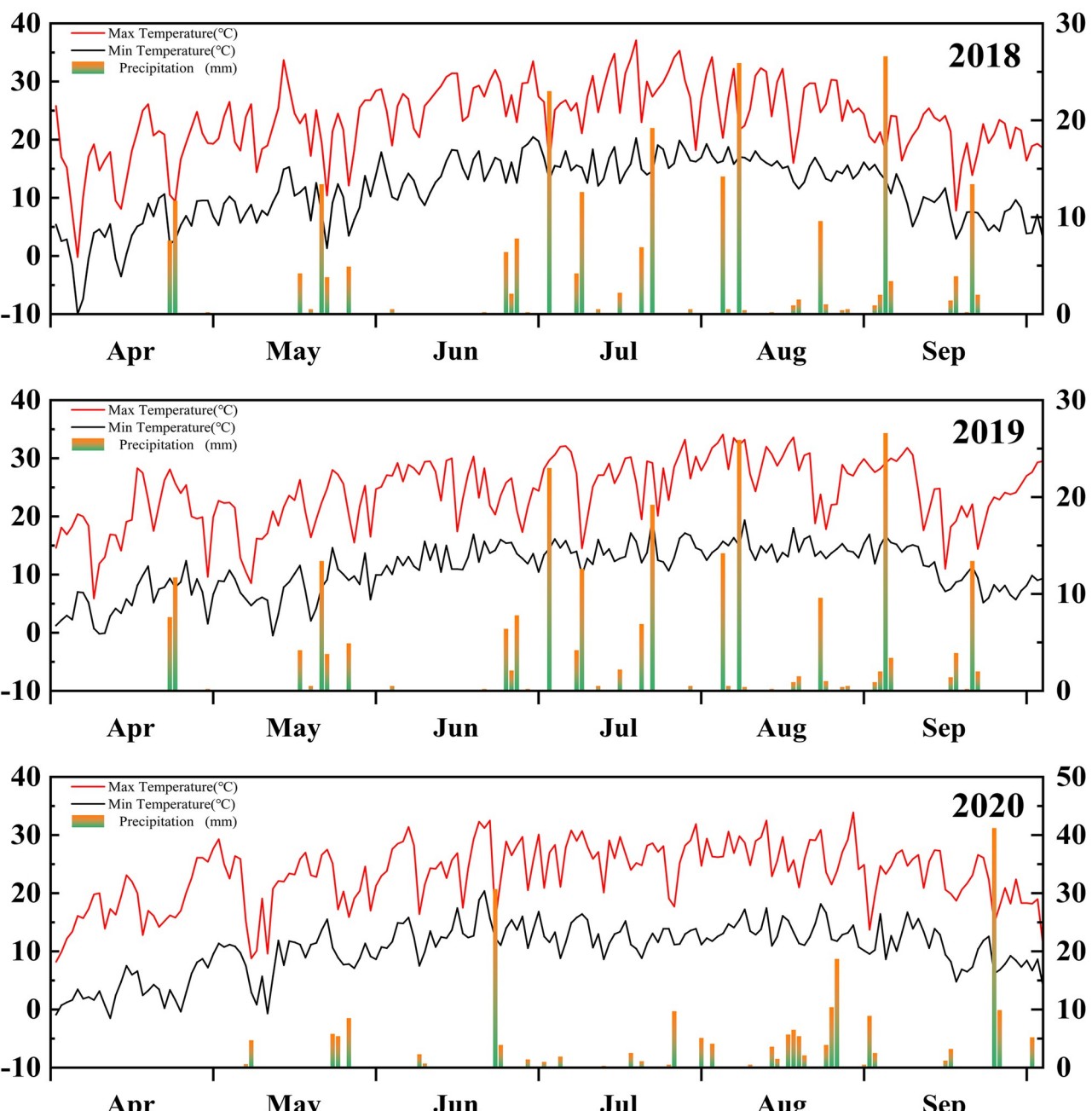

**Fig 1. The precipitations (indicated by the bars) and the max and min daily temperatures (indicated by the curves) of the experimental sites in the growing season of the silage corn in 2018, 2019, 2020.**

exert pressure on the water supply system, and the pressure gauges were connected to the front of the pipes to monitor water pressure inside the pipe.

To identify the stages of crop development, a standardized corn developmental staging system was used [42]. The date was recorded when more than 50% of the plants in each plot reached the vegetative stage (VS) and reproductive stage (RS). The VS stage includes sowing time (ST), germination and emergence (VE), V3 (third leaf), V6 (sixth leaf), V9 (ninth leaf),

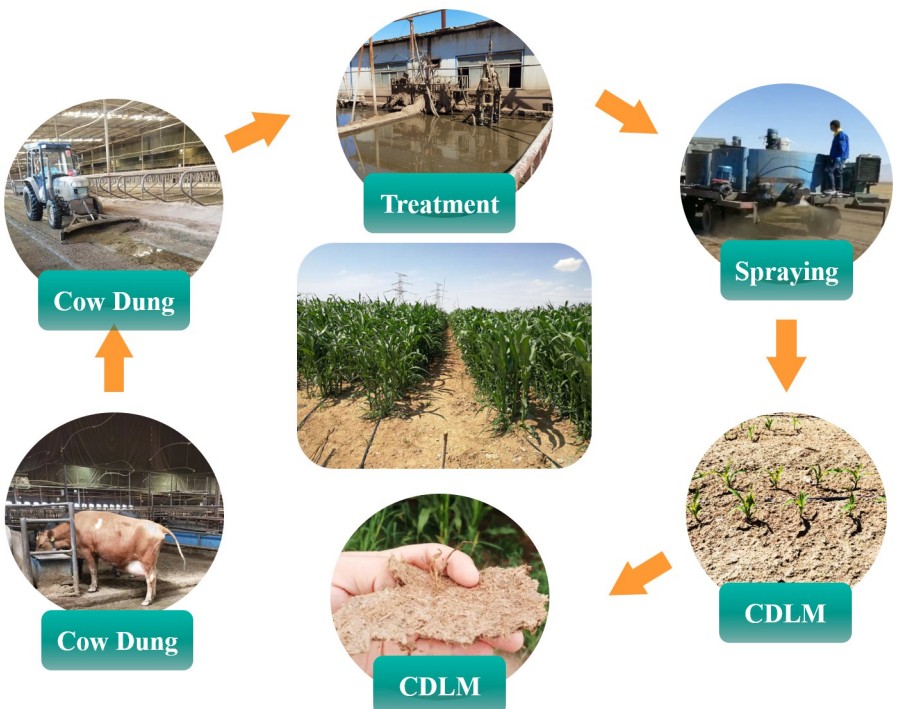

**Fig 2. The process of preparation and spraying of CDLM.**

V12 (twelfth leaf), V15 (fifteenth leaf), and VT (tasseling). The RS stage includes R1 (Silking), R2 (Blister, 10~14 days after Silking), R3 (Milk, 18~22 days after Silking), R4 (Dough, 24–28 days after Silking), R5 (Dent, 35–42 days after Silking), R6 (Physiological maturity, 55–65 days after Silking).

## Sampling measurements

**The degradation of mulching films.** To observe the degradation of mulching films, the pieces of CDLM and PE films with a size of 20cm×20cm were collected randomly during the

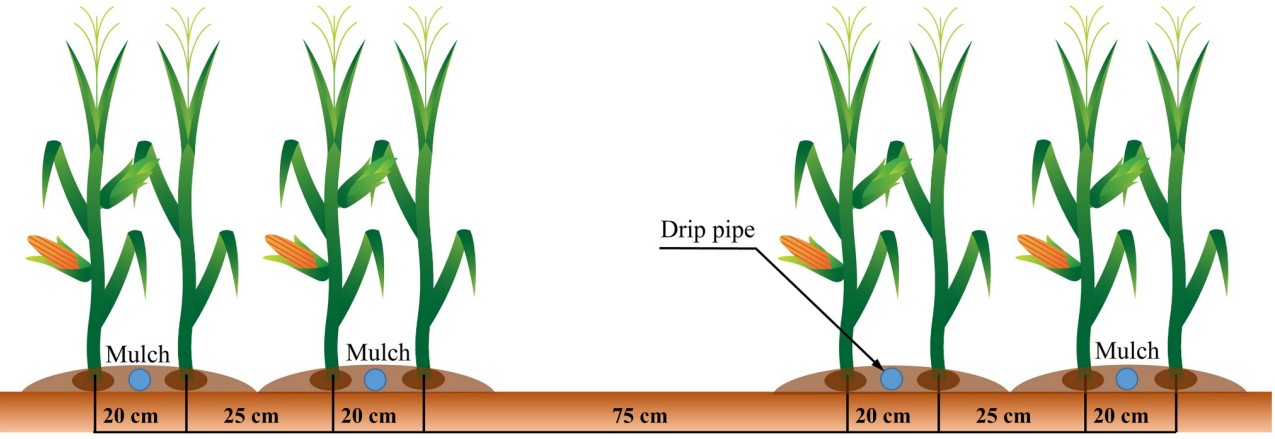

**Fig 3. The narrow/wide row alternation for silage corn in experimental plots.**

**Table 1. Detailed information on irrigation scheduling at different silage corn growth stages used per ha.**

| Growth Stage | Fertilization rule (kg) | | |
|---|---|---|---|
| | $CO(NH_2)_2$ | $CaP_2H_4O_8$ | KCl |
| V3 | 83.3 | 16.67 | 33.33 |
| V6 | 166.67 | 33.33 | 33.33 |
| V10 | 166.67 | 50 | 133.33 |
| V14 | 166.67 | 33.33 | 50 |
| V17 | 166.67 | 50 | 83.33 |
| VT | 166.67 | 50 | 66.67 |

ST, VE, V6, V9, and V12 stages of the silage corn growth in 2020 and the image observation method was used. Degradation of mulching films was observed from April 26, 2020 to August 5, 2020.

**The measurement of soil temperature.** Soil temperature was measured at a depth of 10 cm in each plot using Sansheng Smart Agriculture Platform (Zhangjiakou Sansheng Smart Agriculture Technology Co, Ltd, Hebei, China). The soil temperatures of plots covered by CDLM, PE films, and CK were recorded every 1 h during 24 h periods at a certain day of the critical stages of ST, VE, V6, V9 and V12 in 2018, 2019, and 2020.

**The measurement of soil organic matters.** Soil samples were collected by randomly taking six cores on October 15th, 2018, October 20th, 2019, and October 16th, 2020, (after silage corn harvest) from each plot (10 cm depth). The six soil cores were bulked to form a composite soil sample. The samples were then air-dried, grinded and sieved through 2 mm sieves and analyzed further for the measurement of SOM content. The test was carried out by the standards of GB 9834–88, and the mean values were determined for analysis.

**The measurements of the yield of silage corn and WUE.** The silage corn yield was determined by mechanized harvesting of each test plot and converted to yield per unit area (kg•ha$^{-1}$). The WUE (kg•ha$^{-1}$•mm$^{-1}$) was calculated as the ratio between the annual silage corn yield and the total evapotranspiration (ET) over the growing season in each year [44]. Given the limited influence of groundwater on soil water and the absence of surface runoff and deep percolation at the experiment site, ET was calculated as:

$$ET = P + I + \Delta SWS,$$

where $P$ is the effective rainfall (mm), $I$ the irrigation (mm), and $\Delta SWS$ (mm) the difference in soil water storage at soil depth of about 0–40 cm between sowing and harvest.

**Statistical analysis.** The data processing software Origin2020 was used, and the statistical analyses were used SPSS 24.0 when $p < 0.05$.

## Results and discussion

### The degradation of mulching films

Fig 4 shows the degradations of CDLM and PE films at VS of ST, VE, V6, V18, R2 and R4 stages in 2020. As visible from Fig 4, degradation of CDLM was more noticeable compared with PE films. Fig 4a, 4e, 4i, 4m, 4q and 4u are the surface of CDLM, which was approached with air. Fig 4b, 4f, 4j, 4n, 4r and 4v are the back of CDLM, which was contacted with soil. On April 30 (at the ST stage, four days later of CDLM spraying), the moisture in the CDLM evaporated rapidly, and the surface of CDLM hardened and became a whole block gradually. The back of CDLM completely adhered to the soil, and the color of the surface of CDLM was similar to cow dung, as illustrated in Fig 4a and 4b. After 7 days, on May 3 (at the VE stage), as the

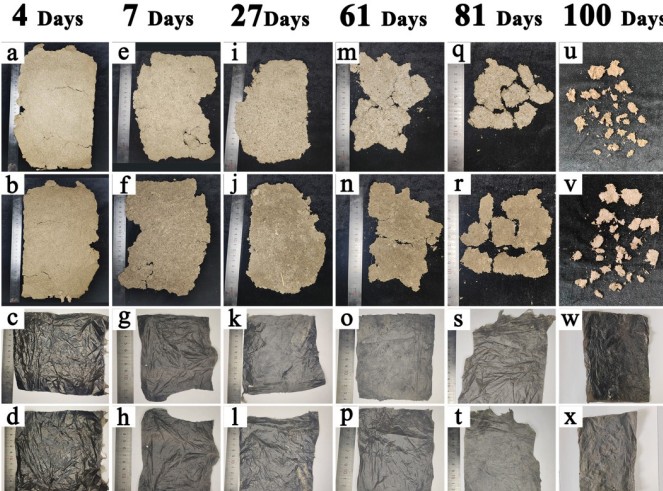

**Fig 4. The degradations of CDLM and PE films at VS of ST, VE, V6, V18, R2 and R4 stages in 2020.**

CDLM hardened further, the combination of the CDLM itself became better than its initial liquid state, making it easier to collect (Fig 4e and 4f). After 27 days, on May 23 (at the V6 stage), the combination of CDLM was the best, and we can collect the complete mulching film at this time (Fig 4i and 4j). After 61 days, on June 27 (at the V9 stage), many cracks appeared, which made it impossible to obtain a sizeable complete film (Fig 4m and 4n). After 81 days, on July 17 (at the V12 stage), the cracks have spread even further, the collection process became more complicated, and some parts of the mulch have merged with the soil (Fig 4q and 4r). After 100 days, on August 5, CDLM could no longer be collected, and CDLM was degraded entirely and fused with the soil (Fig 4u and 4v). These results show that CDLM begins to degrade after 60 days, and it is completely degraded after 100 days. In comparison, observations show that PE films remain unchanged during this period, and the residual mulching films left after the harvest should be recovered to minimize soil contamination. In contrast, the CDLM allows to avoid this operation.

The effects of degradation of CDLM on the silage corn growth were also studied. It showed that, when water evaporated from the CDLM at the ST stage, CDLM hardened rapidly, the back of the side wholly adhered to the soil and formed a covering layer. Because the additional soil was connected to the back of the CDLM during the collection of CDLM at the VE stage, the CDLM became more securely bonded to the soil from the ST stage to the VE stage. Thus, the CDLM could provide heat preservation to the soil and effectively fix soil to prevent erosion from wind and rain, and this effect was carried over into the V6 stage. At the VE stage, the mechanical strength of CDLM was lower than that of PE films, which benefits seedlings breaking through the CDLM, thus guaranteeing emergence rate, as shown in Fig 5a and 5c. In comparison, at the VE stage, the PE films need manual treatments to solve mechanized seeding caused dislocation (between membrane hole and seed hole) to release seedlings. If not, some covered seedings would die when the PE film's strength is too high to penetrate, as shown in Fig 5b and 5d. During the V9 stage, CDLM gradually degraded into fragments, allowing more air and water to diffuse into the soil whereas PE films prevented their penetration (Wang et al., 2020). No changes were observed during V12 stage and CDLM were fused in soil after the V14 stage. These results indicate that the degradation of CDLM is obvious compared to the quasi

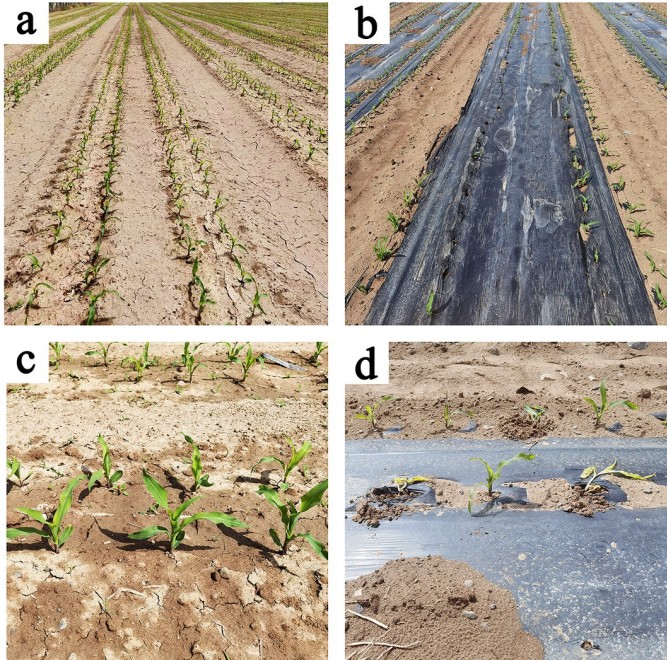

**Fig 5. The silage corn in experimental plots under the CDLM (a and c) and PE (b and d) films at VE stage.**

non-degradation of PE films. The progressive fragmentation of CDLM during crop growth establishes a solid basis for the increase of the yield.

## The measurement of soil temperature

Measurements of outdoor temperature and soil temperatures of plots covered by CDLM, PE films, and CK at a depth of 10 cm are presented in Fig 6.

The initial stage of planting (ST stage) in 2018, 2019, and 2020 are shown in Fig 6a (April 25, normal weather conditions), Fig 6e (April 25, Insolation weather) and Fig 6i (April 25, normal weather conditions) respectively. As shown in Fig 6a and 6i, under normal weather (on April 25, 2018 and April 25, 2020), there was almost no difference in soil temperatures among CDLM, PE films, and CK groups from 4 a.m. to 6 a.m., which is about 5.04˚C on April 25, 2018 and 5.19˚C April 25, 2020. The soil temperature under PE films became higher at noon. The PE films had a maximum temperature 5˚C higher than CK at the same time interval all day long, while CDLM had a temperature of 1~3˚C higher than CK. At 11 p.m., the soil temperatures among CDLM, PE films, and CK groups are almost the same (about 14.03˚C on April 25, 2018 and 13.9˚C April 25, 2020). As illustrated in Fig 6e, when exposed to the sun (on April 23, 2019), the maximum temperature is about 27.9˚C under PE films at 1 p.m. which is higher than the outdoor temperature (about 25.4˚C, at 1 p.m.). Even at midnight, the soil temperature still remained close to 22˚C. while the outdoor temperature is about 18.2˚C, which is a consequence of thermal insulation of PE films. thermal insulation capabilities can be ranked as PE>CDLM>CK after three-year observations. Because of the primary function of mulching films for keeping soil temperature and water to promote the crops germinate in advance, the seeds sown under PE films sprouted 3–6 days earlier than CDLM, whereas seeds sown under CDLM germinated 1–2 days earlier than CK at the same period, as shown in Table 2.

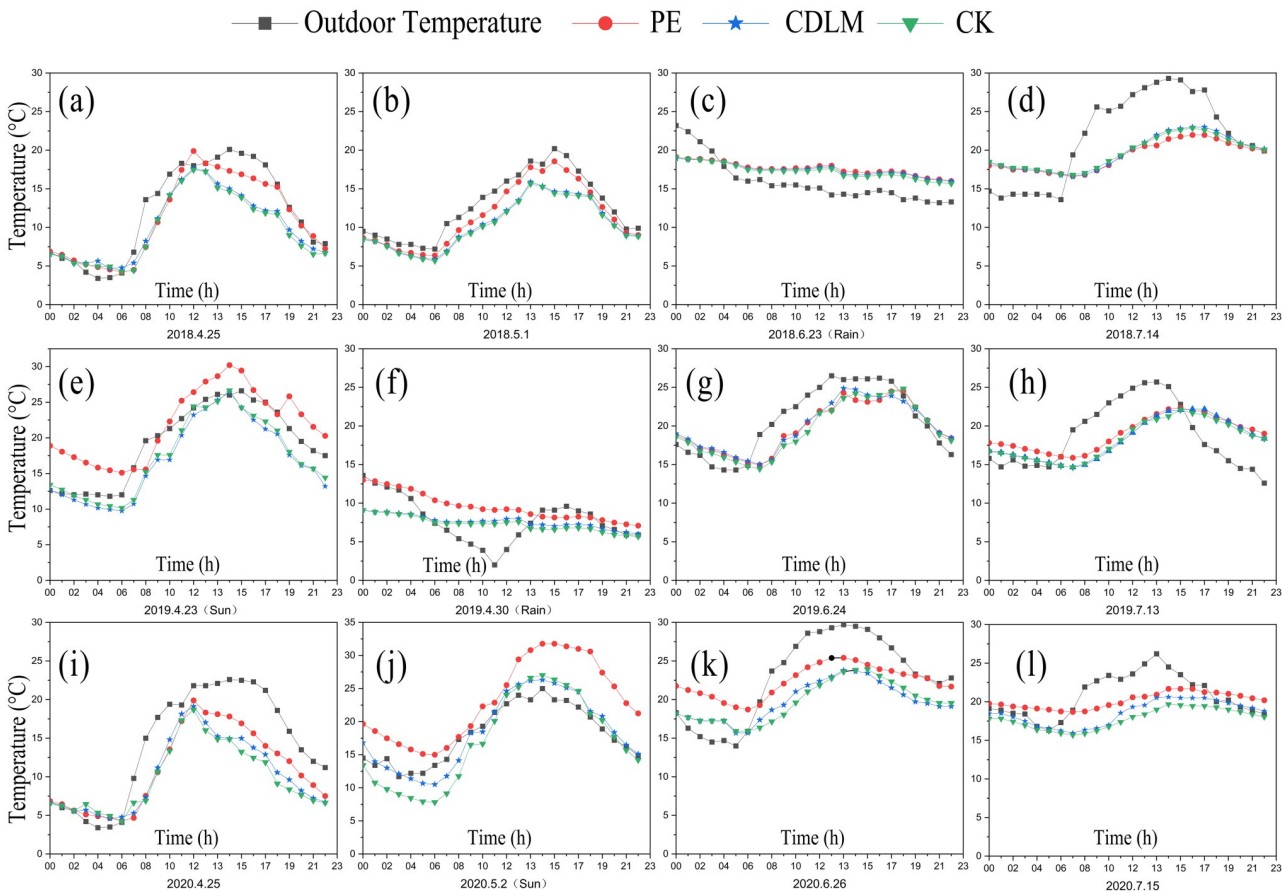

**Fig 6. The outdoor temperature and soil temperatures of plots covered by CDLM, PE films, and CK recorded at every 1 h within 24 h at a certain day in the critical stages of ST, VE, V9 and V12 in 2018, 2019 and 2020.**

Fig 6b, 6f and 6j show the variations of soil temperatures among PE films, CDLM, and CK groups during the VE stage. Under normal weather (on May 1, 2018), the change of soil temperatures in this stage was almost identical to the previous stage. On a rainy day (on April 30, 2019), the soil temperatures under CDLM and CK were below 10°C and remain almost constant over hours. The soil temperature under PE films was not high either, but PE films showed a slightly better thermal insulation. As Fig 6j indicates, in direct sunshine (on May 2, 2020), the soil temperature under PE films was 8°C higher than under the CDLM and CK. During this stage, as the environment gradually gets warmer, the soil temperature rises, so the heat preservation of PE films is secondary. The excessive temperature will prevent the seedling development and even scorch seedlings, affecting the crop development. However, the color of CDLM

**Table 2. The seeding day and the germination day of the silage corn seeds under CDLM, PE films and CK in 2018, 2019, and 2020.**

|  | ST | PE films | CDLM | CK |
|---|---|---|---|---|
|  |  | GT | GT | GT |
| 2018 | 24.Apr | 1.May | 3.May | 6.May |
| 2019 | 23.Apr | 30.Apr | 3.May | 7.May |
| 2020 | 25.Apr | 2.May | 4.May | 7.May |

in this period is darker and can easily absorb sunlight, so the soil temperature under CDLM is 1~2˚C higher than the one of the CK group, which will not cause damages to seedlings.

After entering the V9 stage, the outdoor temperature keeps increasing, at which time the role of mulching films is over. Fig 6c, 6g and 6k show the soil temperatures under CDLM and CK tended to be similar. That is because CDLM begins to degrade, and makes the CDLM better integrated to the soil at this stage. By contrast and regarding degradation, there is no evident change in PE films. As the outdoor temperature rises, the soil temperature under the PE films is higher than CDLM and CK, thus increasing the evaporation of soil water. On the other hand, due to the difference in temperatures between day and night, the surface of PE films can easily cause water accumulation, which also harms crop growth.

In the V12 stage, as shown in Fig 6d, 6h and 6l, the soil temperature under CDLM was nearly identical to that of CK. Because, at this stage, the CDLM has been totally degraded and the crop growth is completed. PE films shaded by the crop are no longer directly exposed to sunlight, the soil temperature under the PE films is close to that of under the CDLM and CK, and occasionally varies depending on outdoor temperature.

According to the three-year observations, the thermal insulation of PE films is superior to that of CDLM. However, the soil temperature under PE films changes rapidly sometimes, which is not conducive to crop growth. In the VE stage, the high temperature even scorches the seedlings and affects the emergence rate. In the V9 stage, the evaporation of soil water and the water accumulation on the surface of PE films limit plant water absorption resulting in yellowing of the leaves [43, 44]. The thermal insulation of CDLM is better than the CK group, and the CDLM can retain heat, enhance the buildup of soil temperature, and aid the seed germination of silage corn.

## The contents of soil organic matters

The contents of soil organic matters under CDLM, PE films and CK after the harvest of silage corn in 2018, 2019 and 2020 were measured. Results are shown in Fig 7. In 2018, the content of soil organic matters was 1.01% in the CDLM, 0.66% in the CK group, and 0.72% in the PE films. In 2019, the content of soil organic matters in the CDLM group was 1.27%, 0.74% in the CK group, and 0.75% in the PE films group. In 2020, the content of soil organic matters in the CDLM group was 1.51%, 0.69% in the CK group, and 0.78% in the PE films group. The content of soil organic matters in the CDLM group increased by nearly 0.3% per year.

Soil organic matter plays a very important role in maintaining soil quality and improving crop yields [45]. PE films improve yields by isolating crops and enabling earlier crop development; however, studies have shown that there is a significant loss of SOM content within 1 to 3 years after mulching due to temperature-induced, accelerated biodegradation [46, 47]. After crop harvest, polyethylene mulch cannot be fully recovery. The residual PE films is difficult to degradation which changes soil structure and make it susceptible to weathering, thus reducing the stability of soil structure and decreasing soil quality [46, 48, 49]. In comparison, residues induced by CDLM degradation, including straw fibers and cow dung, are beneficial to binding the soil generating larger soil aggregates. This extra-SOM plays an important role on macroscopic soil structure stabilization to generate organic clay and mineral complexes as a binding agent at the microscopic level [46, 50].

In this work, the local soils tested were mainly sandy due to their geographical proximity to the Gobi. The liquid film based on cow dung increases the soil organic matter content, improves the soil physicochemical properties and stabilizes the soil structure. It effectively helps crops to absorb other nutrients, especially for lands that are extremely nutrient deficient. The test results showed that the use of CDLM effectively increased the organic matter content

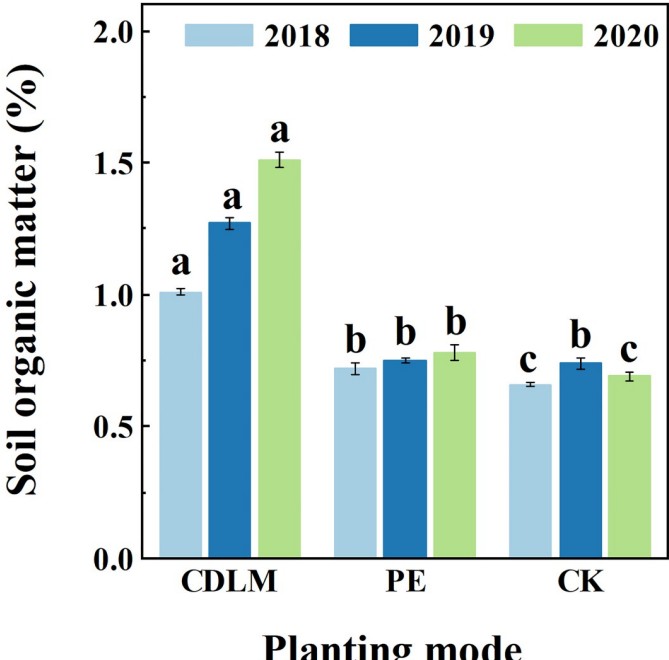

**Fig 7. The contents of soil organic matters under CDLM, PE films and CK after the harvest in 2018, 2019 and 2020.** Different superscripted letters (a, b, and c) indicate significant differences between the treatments in each year (p<0.05).

of the soil, which was twice as high after three years of continuous use as compared to unused soil, while this effect could not be observed on land with PE films.

### Silage corn yield and water use efficiency

The silage corn yields and WUE under CDLM, PE films and CK after harvest in 2018, 2019 and 2020 were measured. Data are shown in Table 3 and Fig 8. The mean silage corn yields under CDLM, PE films and CK over the three years were 91532, 86191, and 78133 kg·ha$^{-1}$, respectively. The three-year experiments showed that the yield of the plot under CDLM was 17.2% higher than that of the CK group, and about 6.2% higher than that of the PE films. The

**Table 3. The silage corn yields and water use efficiency (WUE) under CDLM, PE films and CK in 2018, 2019, and 2020.** Different letters (a, b, and c) within a column indicate significant differences between treatments in a year (p<0.05).

| Growing season | Treatment | Yield (kg·ha$^{-1}$) | ET (mm) | WUE (kg·ha$^{-1}$·mm$^{-1}$) |
|---|---|---|---|---|
| 2018 | CDLM | 91381a | 455 | 201a |
|  | PE | 86036b |  | 189b |
|  | CK | 78093c |  | 171c |
| 2019 | CDLM | 91591a | 453 | 202a |
|  | PE | 86576b |  | 191b |
|  | CK | 77777c |  | 172c |
| 2020 | CDLM | 91621a | 458 | 200a |
|  | PE | 85960b |  | 188b |
|  | CK | 78528c |  | 171c |

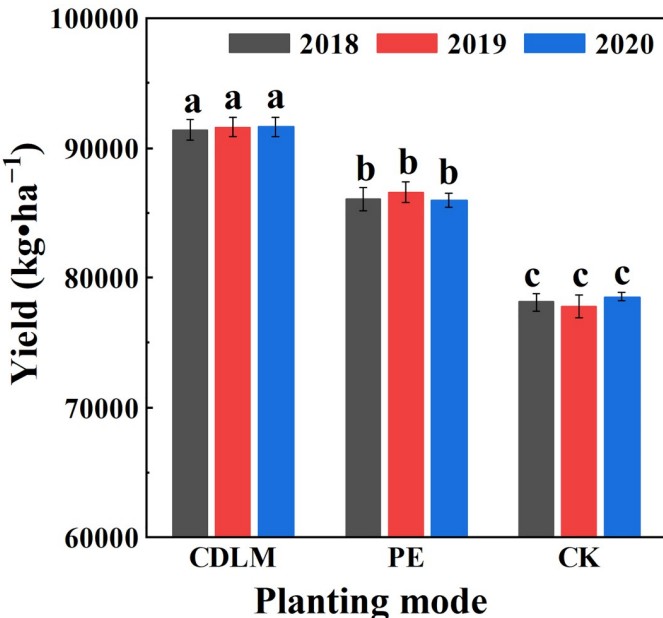

**Fig 8. The silage corn yields under CDLM, PE films and CK in 2018, 2019 and 2020.** Different superscripted letters (a, b, and c) indicate significant differences between the treatments in each year (p<0.05).

mean WUE of the silage corn under CDLM was 200.79 kg•ha$^{-1}$•mm$^{-1}$, which is higher than the one of PE films (189.08 kg•ha$^{-1}$•mm$^{-1}$) and CK (171.40 kg•ha$^{-1}$•mm$^{-1}$).

Film mulching techniques had significantly improved the silage corn yield with respect to no mulching [51, 52]. Through our study, there are three reasons for the crop yield increase. First, although the CDLM covered field presents a little less soil temperature than PE films, the CDLM provides a more satisfactory condition for the initial crop growth than PE films, such as the low strength for seedling breaking and acceptable heat and water preservation effect. Second, the excellent degradation properties of CDLM promote the air injection to maintain soil the soil-air and increase the soil's organic matter, which PE films do not have. Third, the gradual degradation of CDLM increased the contacting of soil and air increasing water utilization on the crop roots when it rains, thus increasing crop yields [44, 53]. At last, after three years of trials, it was shown that the characteristics of CDLM would further enhance the soil environment and finally promoting the growth of the crop and increasing the crop yield.

## Conclusions

In this work, a three-year experiment in a silage corn field was conducted in Hexi Oasis in Gansu to test the potential of CDLM to replace PE films in mulched drip irrigation. The results showed that CDLM had a heat preservation effect and promoted crop germination. The CDLM would degrade completely in ~100 days, and increase 1.01% soil organic matter in three years. Moreover, comparing to PE films, the relatively low strength is a benefit for the breaking of seedlings. The CDLM films were effective in degradation, which would reduce soil pollution generated by PE films and make a cleaner and ecological use of animal husbandry waste. The field experiment results present that the yield laid by CDLM is 6.2% and 17.2% higher than that of PE films and CK, respectively. Therefore, as an alternative to PE films, CDLM has the potential to solve the problem of waste disposal and "white pollution".

## Supporting information

**S1 Table. The precipitations and the max and min daily temperatures of the experimental sites (2018–2020).**
(XLSX)

**S2 Table. Detailed information on irrigation scheduling at different silage corn growth stages.**
(XLSX)

**S3 Table. The outdoor temperature and soil temperatures of plots covered by CDLM, PE films, and CK recorded at every 1 h within 24 h.**
(XLSX)

**S4 Table. The seeding day and the germination day of the silage corn seeds under CDLM, PE films and CK in 2018, 2019, and 2020.**
(XLSX)

**S5 Table. The contents of soil organic matters under CDLM, PE films and CK after the harvest in 2018, 2019 and 2020.**
(XLSX)

**S6 Table. The silage corn yields under CDLM, PE films and CK in 2018, 2019 and 2020.**
(XLSX)

**S7 Table. The silage corn yields and water use efficiency (WUE) under CDLM, PE films and CK in 2018, 2019, and 2020.**
(XLSX)

## Author Contributions

**Conceptualization:** Xiangjun Yang, Lu Li, Wuyun Zhao.

**Data curation:** Xiangjun Yang, Lu Li, Wuyun Zhao, Yongsong Mu, Maohan Chen.

**Formal analysis:** Xiangjun Yang, Lu Li.

**Funding acquisition:** Xiangjun Yang, Wuyun Zhao.

**Investigation:** Xiangjun Yang, Lu Li.

**Methodology:** Xiangjun Yang.

**Project administration:** Xiangjun Yang, Lu Li.

**Resources:** Xiangjun Yang, Lu Li.

**Software:** Xiangjun Yang.

**Supervision:** Xiangjun Yang, Lu Li, Wuyun Zhao, Xuan Li.

**Validation:** Xiangjun Yang.

**Visualization:** Xiangjun Yang, Xuan Li.

**Writing – original draft:** Xiangjun Yang.

**Writing – review & editing:** Xiangjun Yang, Lu Li, Xiaoqiang Wu.

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
