## [Decision Letter · Decision Letter 0]

12 Apr 2022

PONE-D-21-39586Substitute for polyethylene (PE) films: a novel cow dung-based liquid mulch on silage cornfieldsPLOS ONE

Dear Dr. Zhao,

Thank you for submitting your manuscript to PLOS ONE. After careful consideration, we feel that it has merit but does not fully meet PLOS ONE’s publication criteria as it currently stands. Therefore, we invite you to submit a revised version of the manuscript that addresses the points raised during the review process.

We look forward to receiving your revised manuscript.

Kind regards,

Raghvendra Bohara

Academic Editor

PLOS ONE

Journal Requirements:

“Lu Li acknowledges the supports by the Gansu Province Natural Science Foundation for the Youth (20JR10RA554) and GAU-KYQD-2019-16.”

Reviewers' comments:

Reviewer's Responses to Questions

**Comments to the Author**

1. Is the manuscript technically sound, and do the data support the conclusions?

Reviewer #1: Partly

2. Has the statistical analysis been performed appropriately and rigorously? 

Reviewer #1: No

3. Have the authors made all data underlying the findings in their manuscript fully available?

Reviewer #1: No

4. Is the manuscript presented in an intelligible fashion and written in standard English?

Reviewer #1: No

5. Review Comments to the Author

Reviewer #1: RE: PONE-D-21-39586

Based on technical ground (given below & annotated manuscript-attached) and writing rigor the manuscript is not acceptable in current shape.

General comments

The abstract is verbal description of the studies, it needs to be more quantitative, also the test crop, and its management is not been provided in the abstract

The introduction section lacks objective/hypothesis, and has blended with some results particularly in the last paragraph, which is supposed to be a justification and have objectives/hypothesis

Fig, 2- the average temperature is misleading please provide the max and min temperature of the exp location

Fig 3, no need of provision fig 3b and 3c

In methodology section, no crop husbandry procedure is not provided, how much was the seed rate, when was sown, how it was sown, sowing time/methods etc

Table 1, the fertilization treatment is not clear, please change it to hectare base for clarity, also the calculation shows the N, P, K or the sources been used

Table 2 is not need as it contains no significant information

Procedure for determination of the various parameters is not provided or lack some key information. Since the experiment was not been conducted according to any experimental design, lack of replication and randomization make the analysis impossible, then how the experimental data was analysed, and means were compared using prob value of 5%. This section should be carefully revised.

Fig 5, what is surface ad back, not clear, also please proved the scale in days on the a-axis and add pictorial diagram after 100 days as well

Fig 7,8, how these letters have been assigned, which statistical test was been used for this

Table 3, no need of decimal points for the yield and WUE

The results are not according the standard journal manuscript, has described only what has been provided in the table/figure

I donot see any discussion, mechanism for improvement or decline in observed parameter in response to treatment used.

Most of the figure/tables are not needed,

Specific comments

Please see the annotated manuscript attached

6. PLOS authors have the option to publish the peer review history of their article (what does this mean?). If published, this will include your full peer review and any attached files.

Reviewer #1: **Yes: **Ahmad Khan

---

## [Author Response · Author response to Decision Letter 0]

24 May 2022

We answered all of the comments point to point and summited it as ‘Response to Reviewers‘files.

---

## [Editor Report · Decision Letter 1]

28 Jun 2022

Substitute for polyethylene (PE) films: a novel cow dung-based liquid mulch on silage cornfields

PONE-D-21-39586R1

Dear Dr. Zhao,

We’re pleased to inform you that your manuscript has been judged scientifically suitable for publication and will be formally accepted for publication once it meets all outstanding technical requirements.

Kind regards,

Raghvendra Bohara

Academic Editor

PLOS ONE

Additional Editor Comments (optional):

The article can be accepted in the present form- Authors have addressed all the reviewers and my concern
---

## [Editor Report · Acceptance letter]

4 Jul 2022

PONE-D-21-39586R1 

Substitute for polyethylene (PE) films: a novel cow dung-based liquid mulch on silage cornfields 

Dear Dr. Zhao:

I'm pleased to inform you that your manuscript has been deemed suitable for publication in PLOS ONE. Congratulations! Your manuscript is now with our production department. 

Kind regards, 

on behalf of

Dr. Raghvendra Bohara 

Academic Editor

PLOS ONE